# Training Plug-and-Play Knowledge Modules with Deep Context Distillation

**Lucas Caccia**[*]
Microsoft Research Montréal

**Alan Ansell**[*]
University of Cambridge

**Edoardo Ponti**
University of Edinburgh

**Ivan Vulić**
University of Cambridge

**Alessandro Sordoni**
Microsoft Research Montréal

## Abstract

Dynamically integrating new or rapidly evolving information after (Large) Language Model pre-training remains challenging, particularly in low-data scenarios or when dealing with private and specialized documents. In-context learning and retrieval-augmented generation (RAG) face limitations, including their high inference costs and their inability to capture global document information. In this paper, we propose a way of modularizing knowledge by training document-level Knowledge Modules (KMs). KMs are lightweight components implemented as parameter-efficient LoRA modules, which are trained to store information about new documents and can be easily plugged into models on demand. We show that next-token prediction performs poorly as the training objective for KMs. We instead propose Deep Context Distillation: we learn KMs parameters such as to simulate hidden states and logits of a teacher that takes the document in context. Our method outperforms standard next-token prediction and pre-instruction training techniques, across two datasets. Finally, we highlight synergies between KMs and RAG.

## 1 Introduction

Pre-training large language models (LLMs) on massive corpora has shown to be extremely effective at capturing a wealth of general-purpose linguistic and factual knowledge in their parameters. Adapting these models to incorporate new or rapidly evolving information remains challenging, particularly in scenarios where private or specialized documents must be integrated post-hoc. This line of research is most compelling when using LLMs in common enterprise scenarios when proprietary documents often contain the latest instructions, policies, or product details; another scenario is supporting scientific discovery, where LLMs could potentially propose new hypotheses or experiments if they can ingest cutting-edge scientific publications. We need a method that preserves the model's broad capabilities while efficiently encoding novel documents in low-data conditions. Another requirement is for this knowledge to be integrated on demand, in a plug-and-play fashion.

A standard solution to these problems is *in-context learning*, wherein new, up-to-date information is provided in the context of the LLM, before the input prompt. Although many efforts have been devoted to improving long-context models, this approach hits its limit in cases where documents are extremely long. Retrieval-augmented generation (RAG) partially addresses these limitations by only selecting the most relevant passages for a given prompt (Lewis et al., 2020). However, this comes at the cost of: 1) lacking global document information, i.e. only local, or passage-level information is presented in the context, and 2) increased inference costs (in terms of memory footprint and latency) due to the context enlargement. Another alternative to capture global information is to *continually pre-train* the LLM on new documents (e.g., via parameter-efficient methods like LoRA); however,

---

[*]equal contribution.

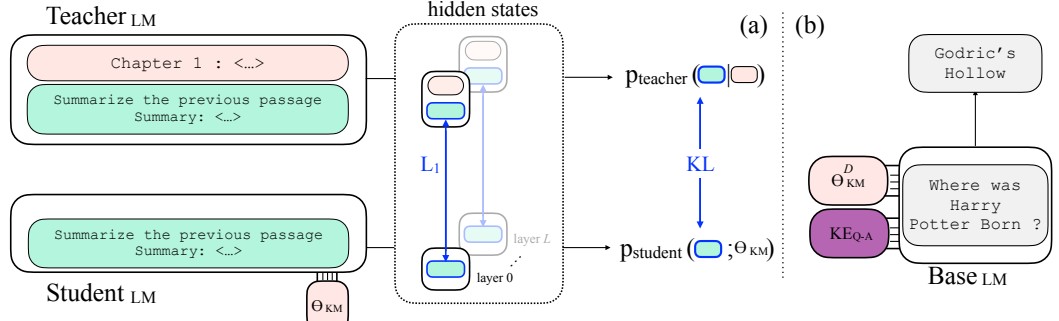

Figure 1: *Left:* Overview of the Deep Context Distillation (DCD) loss. The teacher model conditions on the document to predict (potentially synthetic) relevant data. The student attempts to match both hidden states and logits of this data *without* the document, relying solely on the Knowledge Module (KM). *Right:* Task specific Knowledge Extractors (KE) can be trained and combined with document-level KMs in a *zero-shot* manner.

although next-token prediction is extremely effective during pre-training, it might not be effective in low-data scenarios, as recent literature suggests (Jiang et al., 2024).

In this paper, we aim to train specialized *knowledge modules* (KMs) that encode information about the new document into continuous parameters. Encoding documents into continuous parameter is not new and traces back to Doc2Vec (Le & Mikolov, 2014) and more recently (Zhang et al., 2023a). Here, we parameterize KMs as LoRA modules; this allows for dynamically loading new knowledge on demand and aligns with recent approaches to building more modular LLMs (Ostapenko et al., 2024; Muqeeth et al., 2024). Having one module per document (or per group of documents) enables fine-grained manipulation of information. User-based access control of documents is easily implemented, with document-level KMs visible only to authorized users. When information must be removed from the system, one can simply delete the KM concerned, without impacting other documents or requiring retraining (Fleshman et al., 2024).

Training KMs with next-token prediction is challenging due to data scarcity. New documents contain orders of magnitude fewer tokens compared to pre-training. We enhance the learning signal through two synergistic techniques: *knowledge distillation* and *synthetic data generation*. With distillation, we optimize the parameter of each KM to reproduce the "behavior" of the LLM when it is presented with the new document in context. This can be seen as a generalization of Context Distillation (Snell et al., 2023). We conduct distillation from both the output probabilities and the hidden states, similar to (Sanh et al., 2019). We dub this method "Deep Context Distillation" (DCD). Through synthetic data generation, the model can infer additional facts or relationship entities, which can be seen as intermediate reasoning steps. One of the remaining questions is which inputs we should choose to elicit the behavior of the context-conditioned model. We ablate different variants such as generating summaries, question-answer pairs and recently proposed knowledge graph data (Yang et al., 2025).

Concretely, our contributions are as follows:

- We introduce Knowledge Modules trained with Deep Context Distillation, which effectively condenses document knowledge into a plug-and-play parameter-efficient adapter.
- We show that DCD pairs well with synthetic data generation, with increasing performance as we scale the data generation compute.
- We evaluate our method on two long-context question answering datasets (QuAL-ITY and NarrativeQA) and two base models (Phi-3 3B and Llama-3.1 8B) in two settings: open book and closed book evaluation. In all settings, DCD-based Knowledge Modules outperform all other methods, also displaying synergies with RAG.

## 2 Knowledge Modules

The goal of knowledge modules is to encode information about a particular document $D$ into a small set of parameters such that they can be plugged on top of a base model on demand at inference time.

### 2.1 Parametrization

We parameterize KMs as parameter-efficient adapters using LoRA (Hu et al., 2022). For every linear layer of the base model, LoRA modifies the computation by adding the outer product of two low-rank learnable matrices $A$ and $B$. LoRA adapters have been used in modular approaches due to their ease of batching and merging (Ostapenko et al., 2024; Muqeeth et al., 2024).

### 2.2 Next-Token Prediction Loss

A straightforward way of learning a KM for a new document $D = \{d_1, \ldots, d_N\}$, where $d_t$ is a token, is to use a language modeling (LM) loss such as next-token prediction. Let's call $\theta_{\mathsf{KM}}^D$ the parameters of the KM for document $D$, comprising $A, B$ matrices for all linear layers in the model. The LM loss trains $\theta_{\mathsf{KM}}^D$ to be useful to predict the next token in the document:

$$\mathcal{L}_{LM} = -\sum_i \log p(d_t | d_{<t}; \theta_{\mathsf{KM}}^D), \tag{1}$$

omitting the base model parameters. We will drop the superscript $D$ from the KM parameters for readability. Many variants of this loss have been used in the past both in the original Doc2Vec (Le & Mikolov, 2014) and more recently as span prediction (Xiao et al., 2023). While next-token prediction is the de-facto approach to knowledge extraction when large amount of data is available, it suffers from the *perplexity curse* (Jiang et al., 2024), where the LLM quickly overfits and minimizes the perplexity on the next token, but fails to correctly extract all the relevant knowledge (Berglund et al., 2024).

### 2.3 Deep Context Distillation

In this work, we propose a more effective approach to learn KMs, by relying on distillation (Hinton, 2015), where the model's output is trained to match the output of a teacher network that can have access to additional information or knowledge. Our idea is based on Context Distillation (Snell et al., 2023), originally proposed to internalize task instructions or reasoning steps into a LM. We propose to distill the behavior of a teacher model that has access to document $D$ into a student model that doesn't have access to $D$ but can optimize the parameters of the KM $\theta_{\mathsf{KM}}$. In this way, we hope that $\theta_{\mathsf{KM}}$ will encode the knowledge supporting powerful in-context inferences about $D$. In particular, we perform distillation both in the output (probability) space and the hidden space of the teacher LM, dubbing our loss Deep Context Distillation (DCD). A general form of DCD can be written as follows:

$$\mathcal{L}_{DCD} = \min_{\theta_{\mathsf{KM}}} KL\big(p(\tilde{D}|D) \,||\, p(\tilde{D}; \theta_{\mathsf{KM}})\big) + \sum_l \frac{1}{Z^l} \|h_{\tilde{D}|D}^l - h_{\tilde{D}; \theta_{\mathsf{KM}}}^l\|_1, \tag{2}$$

where $\tilde{D}$ denotes data over which the distillation loss is computed, $p(\tilde{D}|D)$ is the teacher, and $p(\tilde{D}; \theta_{\mathsf{KM}})$ is the student that doesn't have access to $D$. $h_{\tilde{D}|D}^l$ and $h_{\tilde{D}; \theta_{\mathsf{KM}}}^l$ denote the hidden states of the base LM at layer $l$ for the teacher and student respectively, and $Z^l = \|h_{\tilde{D}|D}^l\|_1$ a normalization factor that depends on the $L_1$ norm of the target hidden states. The distillation at the output layer provides a rich training signal as the full teacher distribution is available to the student, while the distillation in the hidden layer provides more direct credit assignment to every LoRA layer. A similar hidden state loss has been also used in DistilBERT (Sanh et al., 2019), albeit the authors using cosine similarity instead of $L_1$. We found our loss to perform well in practice.

A central element left to define to make DCD work in practice is the data $\tilde{D}$ used to perform the distillation process. We propose two versions of our context-distillation loss: the first is document DCD (*DDCD*) and the second is synthetic DCD (*SDCD*).

**Document DCD** In *DDCD*, we sample random chunks of $N$ tokens from the document $D$, and split it in two. With $C_k$ and $C_{k+1}$ denoting these two contiguous chunks, we obtain

$$\mathcal{L}_{DCD} = \min_{\theta_{\mathsf{KM}}} \ KL\big(p(C_{k+1}|C_k) \ || \ p(C_{k+1};\theta_{\mathsf{KM}})\big) + \sum_l \frac{1}{Z^l}\|h^l_{C_{k+1}|C_k} - h^l_{C_{k+1};\theta_{\mathsf{KM}}}\|_1. \tag{3}$$

This approach allows us to leverage the available document as-is, without requiring any augmented data to perform the distillation step.

**Synthetic DCD** In *SDCD*, we instead allow ourselves to generate synthetic data from the document $D$. To do so, we sample a chunk of $N$ tokens from document $C$, $C_k$ and ask the base LM to create synthetic data $S_k$ pertaining to the specific chunk. Then, we minimize:

$$\mathcal{L}_{\mathsf{SDCD}} = \min_{\theta_{\mathsf{KM}}} \ KL\big(p(S_k|C_k) \ || \ p(S_k;\theta_{\mathsf{KM}})\big) + \sum_l \frac{1}{Z^l}\|h^l_{S_k|C_k} - h^l_{S_k;\theta_{\mathsf{KM}}}\|_1. \tag{4}$$

To instantiate $S_k$, we first experiment with question-answer pair generation. As the teacher focuses attends the region of the document containing the answer, the student must encode this information inside the KM. Next, we experiment with summary generation. The idea behind generating summaries is that summaries are likely to require a certain amount of inference about the information contained in the document and therefore this might in turn help encode more information into the KMs by providing a richer training signal. Finally, we also explore the use of Entigraph (Yang et al., 2025), which queries the generator to extract entities from documents and reason about their relationship to one another. Additional information on the synthetic generation process can be found in App. A.1.

### 2.4 Task Adaptation with Knowledge Extractors

The trained KM $\theta_{\mathsf{KM}}$ is task-agnostic, as it is solely trained to reproduce the behavior of the teacher model when $D$ is in context. When some supervised task data is available, we might want to train additional parameters to use the knowledge stored in the KMs to maximize task performance. To do so, we train a Knowledge Extractor (KE), $\theta_{\mathsf{KE}}$, a parameter-efficient module (LoRA) that can combine with the document-specific KMs to maximize task performance. For example, in the context of Question-Answering studied in this paper, if we have available a dataset of questions, answers and supporting documents $\{(q_i, a_i, D_i)\}$, we can train $\theta_{\mathsf{KE}}$ to minimize the negative log-likelihood of the answers when it is combined with every document KM trained separately:

$$\mathcal{L}_{KM+KE} = \min_{\theta_{\mathsf{KE}},w} - \sum_i \log p(a_i|q_i; [\theta^{D_i}_{\mathsf{KM}}, \theta_{\mathsf{KE}}]_w), \tag{5}$$

where $[.]$ denotes our combination function. To combine $\theta_{\mathsf{KE}}$ and $\theta^{D_i}_{\mathsf{KM}}$, we use a learnable weighted combination of the two LoRAs applied after the outer product: $[\theta^{D_i}_{\mathsf{KM}}, \theta_{\mathsf{KE}}]_w = w_M A_{D_i} B^T_{D_i} + w_E A_E B^T_E$, where $w_M, w_E$ are the scalar weights given to the KM and KE parameters respectively. We learn different combination parameters for every layer where LoRA is applied. We will show in the experiments that, although the KE is trained in tandem with a set of documents during training, it can generalize well to extracting knowledge from an unseen set of documents and corresponding KMs in the held-out set.

To also experiment with the synergy between KMs and retrieval-augmented generation approaches, we also extend Eq. 5, to the case where contextual information about the document is included in the context. The extension of the loss is straightforward:

$$\mathcal{L}_{RAG+KM+KE} = \min_{\theta_{\mathsf{KE}},w} - \sum_i \log p(a_i|q_i, P^i_1, \dots, P^i_M, [\theta^{D_i}_{\mathsf{KM}}, \theta_{\mathsf{KE}}]_w), \tag{6}$$

where $P^i_1, \dots, P^i_M$ are document passages retrieved conditioned on query $q_i$.

## 2.5 Training Considerations

Training knowledge modules on a set of documents $\mathcal{D} = \{D_i\}$ can be done in an embarassingly parallel fashion. In contrast, training a KE requires having the set of KMs available for joint multi-task training. KMs can be seen as akin to a continuous indexing mechanism that aims to compress information about a document during training time (similar to GraphRAG (Edge et al., 2024)). Therefore, we operate with the assumption that some of the computation happening at query time can be moved into a more powerful index mechanism at training time. Training KMs requires gradient descent. In the future, efficient ways of initialize KMs to decrease computation time during indexing might be envisioned. $S$DCDalso requires generating summaries from the document. This can be done efficiently with serving libraries such as VLLM (Kwon et al., 2023).

## 3 Experiments

**Setup** We experiment with two question-answering datasets QuALITY (Pang et al., 2022) and NarrativeQA (Kočiskỳ et al., 2018), and two base models, Phi-3 3B[1] (Abdin et al., 2024) and Llama-3.1 8B[2] (Grattafiori et al., 2024) . Unless stated otherwise, we use the intstruction-tuned versions of the model to train KMs. QuALITY is a multi-choice question-answering dataset consisting of 150 training documents and 115 valid documents. The answers of the test documents are private, therefore we report the results on the dev set. The dataset has a variable number of 4-way multiple-choice questions for every document. The average length of documents in the dataset is ∼5,000 tokens. NarrativeQA is a question-answering dataset where documents are longer, ∼60,000 tokens on average. The dataset has 1,102 training documents, 115 dev documents and 355 test documents. For each document, the question-answer pairs in the dataset are created based on a human-written summary of the full document but the evaluation is performed only on the basis of the original documents. Therefore, this dataset is especially challenging for models with limited context length as the gold answers might require assembling information across the whole document. For NarrativeQA, we evaluate performance using multi-reference Rouge-L with the gold answer, while for QuALITY we use Accuracy (25% being random performance).

We experiment with different setups based on the amount of information available to the models at test time. In the *closed book* setting, models do not have access to the document in context while answering the questions, therefore all the models are tested on the basis of how well they can incorporate information before solving the specific task at hand. In the *open book* setting, the document is provided in the context of the models and therefore context processing can be potentially conditioned on the question such as in RAG.

**Methods** In closed book evaluation, we experiment with different ways of training KMs. The first is using the standard LM loss $KM_{LM}$, akin to continual pre-training. Then, we apply our document DCD loss $KM_{DCD}$. We proceed to benchmark our $KM_{SDCD}$, which uses either generated summaries, QA pairs or both as target for deep context distillation. Finally, we experiment with Pre-Instruction Tuning (PIT) (Jiang et al., 2024), where they propose to concatenate task data (query/answer pairs) before the documents during continual pre-training. We denote this variant as $KM_{PIT}$.

In the open book setting, we use RAG and ICL as baselines; for RAG, we split each document into passages of 256 tokens and use SFR-embedding-2 (Meng et al., 2024) to embed passages and questions to perform retrieval of the top-5 relevant passages as measured by the cosine similarity with each question. Similarly to KM, we assume we know the document the questions relate to and therefore we only retrieve passages from that document (we don't search over the set of all documents in the dataset). We report results of zero-shot RAG (just denoted as RAG) and a fine-tuned version of RAG (RAG + KE), where a KE module (just a LoRA adapter) is fine-tuned on the task with the RAG context. For all methods, KEs are always trained solely on the training documents, and never on the dev/test documents.

---

[1]`microsoft/Phi-3-mini-4k-instruct`
[2]`meta-llama/Llama-3.1-8B-Instruct`

| Method | Phi-3 (3B) | | Llama3.1 (8B) | |
|---|---|---|---|---|
| | NarrativeQA | QuALITY | NarrativeQA | QuALITY |
| *Closed Book* | | | | |
| Base Model | 12.1 | 26.6 | 12.5 | 27.8 |
| $KM_{LM}$ | 15.2 | 26.4 | 17.9 | 27.0 |
| $KM_{DDCD}$ | 21.2 | 28.7 | 24.5 | 31.1 |
| $KM_{PIT}$ | 11.5 | 28.9 | 11.9 | 31.2 |
| $KM_{SDCD}$ | | | | |
| - Summary | 23.9 | 32.5 | 26.6 | 37.2 |
| - QA | 23.5 | 33.3 | 26.0 | 37.3 |
| - Summary + QA | **25.8** | 34.0 | 27.3 | 39.0 |
|   - w/ concat | 25.4 | **34.1** | **28.6** | **39.8** |
| KE | 19.6 | 39.4 | 20.9 | 39.3 |
| $KM_{LM}$ + KE | 20.7 | 36.0 | 22.3 | 40.9 |
| $KM_{DDCD}$ + KE | 28.0 | 40.4 | 31.2 | 45.8 |
| $KM_{SDCD}$ + KE | | | | |
| - Summary | 32.2 | 42.7 | 32.5 | 57.2 |
| - QA | 32.5 | 42.5 | 32.7 | 56.5 |
| - Summary + QA | **34.8** | **45.0** | 33.8 | 59.1 |
|   - w/ concat | 33.7 | 44.3 | **34.7** | **59.3** |
| *Open Book* | | | | |
| ICL | 21.0 | 33.1 | 38.0 | 46.0 |
| RAG | 23.0 | 34.7 | 28.2 | 41.0 |
| RAG + $KM_{SDCD}$ | 23.0 | **35.6** | 26.4 | 41.1 |
|   - w/ concat | **26.2** | 35.5 | **31.2** | **41.6** |
| RAG + KE | 36.2 | 53.4 | 36.7 | 62.2 |
| RAG + $KM_{SDCD}$ + KE | **40.4** | **55.8** | 40.8 | 66.3 |
|   - w/ concat | 39.7 | 55.2 | **41.4** | **67.6** |

Table 1: Results for QuALITY and NarrativeQA on Phi-3 (3B) and Llama3.1 (8B).

We experiment with combinations of RAG and KMs, namely RAG + KM (zero-shot), and RAG + KE + KM (KE trained to combine RAG and KM information) to analyze the synergy between RAG and KMs. Technically, ICL for decoder-only models can be considered as a closed book approach if the KV document cache is stored in memory. However, this comes at an excessive storage cost (for Llama3.1 8B, for a document of 60k tokens, it's ∼30Gb). We experiment with KV compression methods such as the recently proposed L2 compress (Devoto et al., 2024b) to analyze tradeoffs between performance and storage cost.

KMs and KEs methods use LoRA as parameter-efficient adapter with a rank of 16, LoRA alpha of 16, learning rate of 1e-3, are trained with 1500 steps with batch size of 8 and cosine learning rate scheduler. We use greedy decoding to sample the responses for NarrativeQA.

**Concatenating Synthetic Data** In its current form, Knowledge Module training may suffer from a distribution shift when combined with RAG approaches; indeed, since the synthetic data is relatively short, the KM may struggle to adapt in a zero-shot manner to longer RAG contexts. To address this, we propose to concatenate several synthetic data samples stemming from the same document chunk, in order to artificially increase the training context for the student. The rest of the training procedure is kept unchanged.

**Results** We report the results for the two base models and the two datasets in Table 1. Results are consistent and show that *SDCD* performs best across the board in the closed book setting, outperforming both LM, PIT and *DDCD*. The result hold if we train on Summary, QA, or both, the latter performing the best. In the open book setting, we see that KMs struggle at zero-shot combination with RAG (RAG + KM lines). For Llama-3.1, there

| Method | LongHealth | | Synthetic Data Type | LM | +KE | DCD | +KE |
|---|---|---|---|---|---|---|---|
| | | +KE$_{QL}$ | Summary | 32.1 | 49.7 | 38.3 | 54.7 |
| ICL | **79.0** | 79.3 | QA | 33.1 | 53.6 | 39.2 | 59.1 |
| ICL + KM$_{SLM}$ | 65.6 | 78.3 | Entigraph | 35.0 | 55.6 | 39.4 | 55.4 |
| ICL + KM$_{SDCD}$ | 75.3 | **80.3** | Sum. + QA + Entig. | 36.6 | 58.4 | 40.0 | 60.5 |

Table 2: (Left) Results for the LongHealth dataset; (Right) Ablation of different types of synthetic data on QuALITY for training KMs with both LM and $S$DCDlosses.

are slight signs of forgetting (comparing RAG vs. RAG + KM), which are due to the fact that KMs are never trained with long contexts. Indeed, we see that training with longer concatenated samples addresses this gap, showing consistent gains over RAG. Moreover, we see a similar performance boost when training a specialized KE, which highlights strongly synergistic behavior of RAG and KMs. Comparing RAG + KE and RAG + KM + KE, we see **4.2 & 4.1 Rouge-L** gains on NQA and **2.4 & 4.1% Accuracy** improvements on QuALITY, for Phi-3 & Llama 3.1 respectively. Lastly, we note that in settings where no synthetically generated data is available, our distillation objective ($D$DCD) proves to be a better learning signal than next-work prediction (LM). A more thorough analysis of this observation is discussed in B.2.

In Figure 2, we relate performance on NQA vs. token cost at inference time, as measured as the number of context tokens the model consumes to produce an answer. We denote by $k$ the number of retrieved passages for RAG. For this study, we use the KM trained with $S$DCD using Summary information. We see that KM+KE (without any retrieval) outperforms retrieving RAG+KE with $k = 1$ while only using the question tokens in the prompt (40 tokens in average vs 200 tokens for RAG+KE with $k = 1$). Scaling the number of retrieved passages $k$ benefits both RAG + KM + KE and RAG + KE, while introducing KM retains gains over RAG for similar values of $k$ and matches performance at a lower cost (42.4 obtained with $k = 8$ vs 42.5 attained with $k = 16$) providing savings of 50% at inference time.

**Transfer of Knowledge Extractors**  We have shown up until now that Knowledge Extractors can be combined in a zero-shot manner with KMs from new documents, albeit trained on a similar data distribution as the KE. However, in many realistic applications of document understanding, we may not have access to supervised data with which to train a KE. To test whether a Knowledge Extractor trained for Question Answering can be applied synergistically with "out-of-distribution" KMs, we run the following experiments. We leverage the LongHealth dataset (Adams et al., 2024), a collection of clinical reports for 20 fictional patients, totalling roughly 11K tokens per patient. The benchmark contains 20 questions per patients, totalling 400 five-way multiple choice questions. For train Knowledge Modules for each patients following the same procedure as in prior experiments. We then apply zero-shot the Knowledge Extractors trained on QuALITY, with the patient's document in-context. Note that we apply the respective KEs for each method, e.g. the extractor trained for DCD is applied on DCD-trained KMs. Results are shown in Table 2 (left). First, we see that KMs combined zero-shot with standard ICL fail to deliver additional gains. However, we see that using a KE trained on QuALITY can transfer well in a zero-shot "plug-and-play" fashion and achieves 1.3% accuracy improvement over ICL. Overall, this highlights the potential of building modular solutions for knowledge injection, enabling practitioners to easily reuse existing blocks and customize a model.

**Ablations**  Having established that DCD with synthetic data is consistently outperforming other baselines, we investigate the extent to which better, more diverse synthetic data helps performance. To do so, we test our approach by leveraging different types of data sampled from GPT-4o (OpenAI, 2024). In addition to Summary and QA, we also leverage Entigraph (Yang et al., 2025), which prompts the model to extract relevant entities, and how they relate to one another. Preprocessing details can be found in App. A.3. In Table 2, we show the results of this approach. For completeness, we also compare to using next-token

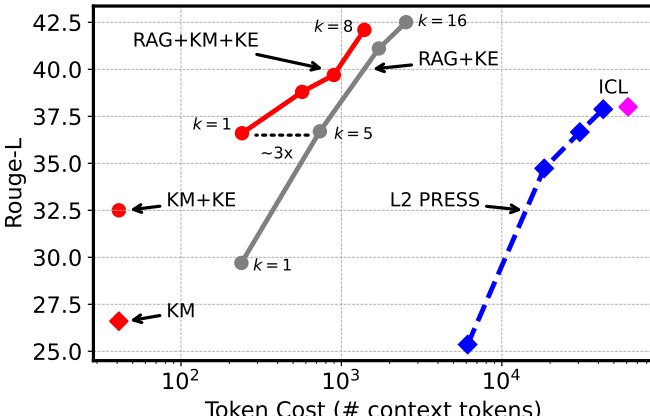

Figure 2: Number of tokens in the context for every model (on the x-axis) vs Rouge-L performance (on the y-axis) on the NarrativeQA dataset. We report both zero-shot (diamond marker) and fine-tuned models (circle marker, KE). We also benchmark a recent KV-cache compression method based on the l2 norm of the key vectors (Devoto et al., 2024b).

prediction (LM) on the same data. We see that our approach clearly benefits from having a diverse set of synthetic data for the distillation step. Moreover, comparing to results in Table 1, we see that for Summary and QA data types, generating from a stronger model yields stronger results.

Next, we explore if the different components of our DCD loss are useful. In Tab. 4, we show that additionally matching the hidden states does provide benefits in settings where there are no KEs. Indeed, combining both KL and $L_1$ losses gives the best performance. When a task-specific KE is available, we see that the KE can mitigate this difference in performance gap. We also experiment with varying the number of synthetic datapoints available per document chunk. In Tab. 5 we show that as we increase the number of synthetically generated data for each chunk, the final performance is increased. Overall, increasing the amount of offline computation in the synthetic generation step, either by generating more data, or from a stronger model, leads to higher performance.

Lastly, we investigate whether applying specialized KMs leads to forgetting of general knowledge. We evaluate Llama 3.1 (8B) augmented with a random Knowledge Module on the multiple-choice MMLU dataset (Hendrycks et al., 2021), used to measure the model's knowledge on a wide variety of topics. We find that performance (accuracy) is only negligibly affected, dropping 1% from **63.9%** to **62.9%**. More details can be found in B.3.

## 4  Related Work

**In-Context Learning and RAG** have been used to incorporate external knowledge at inference time. While powerful, these techniques can impose high storage costs (e.g., storing large key-value caches). Efforts to compress KV caches typically use heuristics such as norms of the key-value pairs (Devoto et al., 2024a) or learned. Recent methods try to build more powerful RAG mechanisms by scaling both indexing and generation (Yue et al., 2024; Edge et al., 2024). Yue et al. (2024) extend RAG by retrieving passages, queries, and answers. They further scale inference time by decomposes queries into sub-queries, incrementally building solutions. GraphRAG Edge et al. (2024) has a costly indexing step which builds a knowledge graph from chunked documents by extracting (subject, predicate, object) relations, then clusters and summarizes sub-graphs. This offline graph-based pre-processing is reminiscent of KMs in reducing repeated computation at inference time. Much like KMs, GraphRAG help capturing global document context. Data generated by GraphRAG at indexing time could potentially be used as distillation data in KMs.

|  | Adapter Granularity | Loss Function | Synthetic Data | Module Composition |
|---|---|---|---|---|
| Xiao et al. (2023) | Document | LM, Masked LM | No | Yes |
| Zhang et al. (2023a) | Corpus | Masked LM | No | Yes |
| Kujanpää et al. (2024)[a] | Corpus | Distillation | Yes | No |
| **Knowledge Modules** | Document | Distillation | Yes | Yes |

[a] Concurrent work

Table 3: Comparison of Knowledge Injection Methods with respect to our proposed approach, Knowledge Modules.

**Knowledge Distillation** typically transfers logits or intermediate representations from a teacher model to a student (Xu et al., 2024). While effective at compressing large models, these approaches usually differ from KMs in that they do not explicitly store domain knowledge for repeated use in specific tasks. Context distillation (Snell et al., 2023) aims to "absorb" a context into a model in a way that preserves its impact on generation. Early work focused on toxicity and safety contexts (Askell et al., 2021) and for internalizing reasoning traces (Snell et al., 2023). Recently, Shin et al. (2024) propose Generative Context Distillation, augmenting the distillation objective with a generative loss to match model outputs more comprehensively. Choi et al. (2023) use generative distillation to internalize long prompts, first training a generator on data sampled from the prompt-conditioned LLM, and subsequently using the generator's samples for the distillation step. In contrast to these approaches, the work presented in this paper goes beyond prompt internalization, and investigates distillation as a mechanism for knowledge acquisition. Recently, Qi et al. (2025) show that a distillation objective can be used for localized knowledge editing given (entity, relation, object) triplets. Finally, concurrent to our work, Kujanpää et al. (2024) also explore the synergy between knowledge distillation and synthetic data generation. Our work still differs by the *modularity* or our solution, which consists in learning a KM for *each* document and demonstrate transferability of the solution across unseen KMs. Finally, our work goes beyond self-distillation, as we show that using synthetic data from a stronger (but off-policy) model yields additional gains. See Tab. 3 for a summary.

**Knowledge Injection with Modules** Several works have similar goals of KMs. Xiao et al. (2023) introduce "plug-and-play" document modules where an MLP transforms encoded document tokens into soft prompts used across attention layers. Unlike the work in Xiao et al. (2023), we require that the inference compute is fixed and does not scale with the length of the document $D$. Zhang et al. (2023a) similarly train knowledge modules in a task-agnostic or task-specific manner, caching the processed document representation for efficient reuse. Amortization-based approaches train a "hypernetwork" to generate lightweight adaptations (e.g., low-rank shifts or prefix-tuning parameters) from a given context. Chen et al. (2024) learn to project context tokens into low-rank parameter shifts in a self-supervised manner, and Tack et al. (2024) encode documents into prefix-tuning weights using a T5-based hypernetwork. These methods train the hypernet on multi-task data, so at inference time it can produce task-specific or document-specific modules in a single forward pass. KMs as described in this paper are trained with gradient descent on single documents independent of any multi-task training. This increases per-document costs but reduces the risk of domain shift. Future work might be devoted to efficient learning of KMs.

**KV Cache Compression** Our work can be seen as an instantiation of context compression (Nawrot et al., 2024). Knowledge Modules operate in the *query-agnostic* setting (Zhang et al., 2023b; Devoto et al., 2024a), where compression is performed without knowledge of the downstream task. Several approaches can also train the context compressor, typically with cross-entropy, or a reconstruction objective (Ge et al., 2024; Qin et al., 2024) More closely related to ours, Chari et al. (2024) leverage a distillation objective to perform prompt compression into a soft-prompt-like adapter.

## 5 Conclusion

In this work, we introduced a novel approach to encode document-specific knowledge into lightweight, plug-and-play Knowledge Modules (KMs) using Deep Context Distillation (DCD). We showed that DCD, especially paired with synthetic data, outperforms traditional next-token prediction and pre-instruction tuning across several question-answering benchmarks. Our results highlight DCD's consistent performance gains, its synergy with retrieval-augmented generation (RAG), and the modularity that enables fine-grained control over knowledge integration.

The current investigation focused on the setting where the document pertaining to the question is known. For future work, we are excited by the potential combination of KMs with zero-shot routing methods (Ostapenko et al., 2024; Muqeeth et al., 2024) to deploy our approach in *document-agnostic* settings, where the system can combine information from multiple KMs.

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

## A  Synthetic Data Generation

Below we provide additional information on the generated data used for our experiments. We split documents into non-overlapping chunks of 2048 tokens, using `RecursiveCharacterTextSplitter` from the `langchain_text_splitters` package. The prompts are provided below.

### A.1  Summary Generations

We generate a maximum of 16 summaries for each chunk. For our main experiments, we use a default of 8 summaries per chunk, unless stated otherwise.

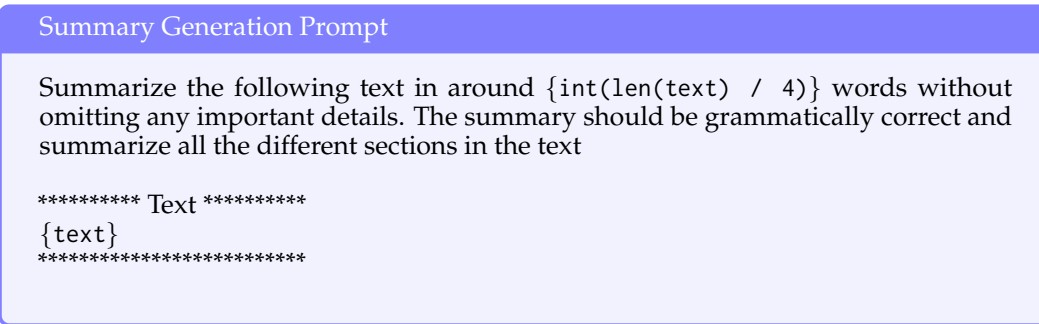

Summary Generation Prompt

Summarize the following text in around {int(len(text) / 4)} words without omitting any important details. The summary should be grammatically correct and summarize all the different sections in the text

********** Text **********
{text}
**************************

## A.2 Q/A Generation

> **Q/A Generation Prompt**
>
> Create `num_questions` questions that can be answerable from the following text, along with their answers. Strive to generate challenging questions that require aggregating information across the provided text. Focus on different sections of the text to increase diversity of the generated questions. Format your answer as follows:
>
> <question id=1 >QUESTION 1 HERE <question >
> <answer id=1 >ANSWER 1 HERE <answer >
> <question id=2 >QUESTION 2 HERE <question >
> <answer id=2 >ANSWER 2 HERE <answer >
>
> ********** Text **********
> {text}
> ************************

## A.3 Entigraph Generation

For the Entigraph data, we use the publicly available Entigraph generated dataset on the QuALITY dataset available on HuggingFace under `zitongyang/entigraph-quality-corpus`. The dataset provised 1 (large) synthetically generated document per document in the original dataset. In order to obtain smaller chunks of synthetic data, we split according to `<|entigraphseptoekn|>`, which gives us chunks of ~700 tokes on average describing a specific relationship between two entities. As we do not know where the information is contained in the original document, we provide the full document in context during training.

# B Ablations

## B.1 Impact of different losses

The following ablation is done on a 140 document subset of NQA (100 train, 20 dev, 20 test).

| Method | Rouge-L (no KE) | Rouge-L (KE) |
|---|---|---|
| KM (Summary DCD) | 23.1 | 32.3 |
| KM (Summary DCD, hidden loss only) | 18.8 | 29.5 |
| KM (Summary DCD, logit loss only) | 21.7 | 32.3 |

Table 4: Ablation on the each distillation component using Phi-3 as the backbone.

| Method | Rouge-L (no KE) | Rouge-L (KE) |
|---|---|---|
| KM (2 summaries) | 20.4 | 31.0 |
| KM (4 summaries) | 21.8 | 31.5 |
| KM (8 summaries) | 23.2 | 31.7 |
| KM (16 summaries) | 24.4 | 32.3 |
| KM (Q/A pairs) | 22.1 | 31.7 |
| KM (summaries & Q/A pairs) | 26.3 | 32.7 |

Table 5: Ablation on the data used for DCD training

## B.2 Impact of Document DCD over standard next-token prediction

In this section we investigate *why* Document DCD outperforms the standard language modeling objective. We hypothetise that this is due to the richer signal that the distillation objective provides. Indeed, both the hidden state matching loss, as well as the KL objective, yield more information to the learner than the standard next-token prediction objective. This result aligns with the general finding that distillation (teacher model matching) is more data-efficient than NTP (data matching) in the literature (He et al., 2022; Minixhofer et al., 2025).

To prove this, we run the following experiment: We progressively strip components of $D$DCD, making it closer to the LM baseline; we remove the l1 loss on the hidden states, and progressively lower the temperature when computing the teacher probabilities $p_{teacher} = \text{softmax}(\frac{logits}{\tau})$, until The target distribution becomes one-hot, effectively mimicking the properties of next-token-prediction loss.

| Model | Rouge-L |
|---|---|
| $D$DCD | 21.2 |
| $D$DCD without hidden state loss | 19.0 |
| $D$DCD without hidden state loss, $\tau = 0.1$ | 16.2 |
| $D$DCD without hidden state loss, $\tau = 0.001$ | 16.0 |
| $D$DCD without hidden state loss, $\tau = 0$ | 15.9 |
| LM baseline | 15.2 |

Table 6: Rouge-L scores for different $D$DCD model variants.

From these results, we can see that the performance correspondingly converges to that of LM. Experiments are run on the NarrativeQA dataset with phi-3 as the base model. Thus, we conclude that the overall distillation objective is central to the observed gains w.r.t LM.

## B.3 Forgetting of General Knowledge after KM training

We believe our approach to be more robust to downsides associated with knowledge injection, given the plug-and-play nature of Knowledge Modules. Indeed, if other information is altered in unintended ways, we always have the opportunity to downweight (or outright remove) the Knowledge Module for a given document. Because we never modify the base model, we can always revert to a previous capability present in the base model. That said, we test how much general knowledge is forgotten after Knowledge Module training, per your suggestion. We evaluate the KMs on the MMLU (Hendrycks et al., 2021) dataset, where for each datapoint, we randomly sample a KM with which to answer the question. We use a subset of 400 examples, evenly split across all MMLU subjects. We compare to the base model, llama 3 8B-Instruct below

| Model | Accuracy |
|---|---|
| base model | 63.9 |
| base model + KM | 62.9 |

Table 7: Accuracy comparison between the base model and its variant with KM.

We see that the base model equipped with a Knowledge Module is able to retain its overall general knowledge, dropping only 1 point on MMLU.

