# OpenReview forum: "Training Plug-and-Play Knowledge Modules with Deep Context Distillation"
_colmweb.org/COLM/2025/Conference — COLM 2025_

### Official Review · Reviewer_1VZd · 2025-05-07

**Rating:** 7
**Confidence:** 3
**Ethics Flag:** 1

**Summary:**

This paper proposes a novel method for integrating new information into Large Language Models (LLMs) called Knowledge Modules (KMs) trained with Deep Context Distillation (DCD). The paper presents a new training objective, Deep Context Distillation (DCD), which aims to train lightweight, parameter-efficient KMs (implemented as LoRA modules) to encode document-level knowledge. The authors demonstrate that DCD, especially when combined with synthetic data generation, outperforms standard next-token prediction and pre-instruction tuning techniques in closed-book settings on two question-answering datasets, QUALITY and NarrativeQA, using Phi-3 3B and Llama-3.1 8B base models.

**Reasons To Accept:**

1. The paper introduces a novel and original approach for dynamically integrating knowledge into LLMs through Knowledge Modules (KMs) trained with Deep Context Distillation (DCD). This method provides a distinct alternative to existing techniques like in-context learning and RAG, particularly excelling in low-data scenarios.

2. The paper provides clear mathematical formulations for the different loss components. The experimental evaluation is thorough, using relevant datasets (QUALITY and NarrativeQA) and appropriate metrics (Rouge-L and Accuracy).

3. The paper addresses a critical challenge in working with LLMs: efficiently updating and managing knowledge, especially with private or rapidly changing information. The proposed plug-and-play nature of KMs offers a practical solution for enterprise and specialized applications where full model retraining is infeasible.

**Reasons To Reject:**

1. The paper highlights the importance of synthetic data for training KMs in low-data settings. However, the sensitivity of the proposed method to the quality and characteristics of this synthetic data is not fully explored.

2. While KMs are presented as lightweight for inference, the training process involves distilling from a teacher model that sees the full document in context. The computational cost and complexity of this distillation process, especially for very long documents or large numbers of documents, could be a concern.

---

> ### Author Response · Authors · 2025-05-31
> **Response to 1VZd**
>
> Thank you for providing a thorough review of our work. We discuss the points raised below.
>
> $\texttt{ }$
> ### On the sensitivity to the quality and characteristics of the synthetic data
>
> We agree that an extensive analysis on the quality of the generated data would be interesting. We tried to assess the impact of different types of synthetic data in Table 2, namely the usage of document summaries, Q/A pairs, and the use of Entigraph [5], a recent synthetic data generation technique that excels in knowledge extraction from documents, and leverages a stronger base model, GPT-4o. In general, we find that generating from a stronger model (GPT-4o) and using more advanced methods like Entigraph does lead to more performant KMs. Moreover, these different types of synthetic data can be combined for additional gains.
>
>
> $\texttt{ }$
> ### On the cost of training KMs
>
>
> We believe KMs to still be a go-to solution for settings where the number of inference queries per document is high (kindly refer to the global response above), or settings where users want to maximize performance over efficiency, where KMs + RAG outperforms other approaches (Table 1, bottom row). Moreover, in other settings where user experience is top of mind, and latency should be low, KMs enable shifting the cost away from the user.
>
>
> Thank you again for providing your feedback!
>
> [5] Yang et al., “Synthetic Continued Pretraining”

---

> > ### Comment · Reviewer_1VZd · 2025-06-06
> >
> > Thanks for the comments. They've solved some of my concerns, and I've raised the scores accordingly.

---

### Official Review · Reviewer_LZaV · 2025-05-12

**Rating:** 7
**Confidence:** 4
**Ethics Flag:** 1

**Summary:**

This paper proposes a knowledge module (KM), which is a LoRA adapter that is trained to mimic the logics of a teacher model that has access to new documents, without directly accessing the new documents. In other words, KM is designed to learn the new information by distilling from a model that has access to these new information, so that a model equiped with KMs no longer needs to access the new documents at inference-time, which saves compute and storage. The authors proposed several training variants of KMs, namely DDCD (which treats new documents as chunks and try to recover chunks given context), SDCD, which distills generated question and answer pairs based on the new documents instead of raw texts, and KMs can be optionally equiped with knowledge extractors (KE) that can further optimize based on task-specific supervision data.

Experiments show that the proposed KM methods outperform continual pre-training baselines, as well as in-context learning and RAG baselines in certain settings.

**Reasons To Accept:**

The proposed knowledge module methods are interesting and easy to use, and they can potentially be applied in many applications that require fast and efficient inferences and deployment of private-domain data. The proposed experiments are comprehensive and demonstrate the effectiveness and versatility of KMs.

**Reasons To Reject:**

1. The relatively lower performance compared with ICL/RAG baselines across models and datasets in unsupervised settings can be concerning. This means that ICL and RAG are still going to be the primary resources for private-data applications, at least for smaller models where the additional computational cost is minimized. Ideally other experiments with larger base LMs may further support the story of this paper as it is more beneifical to apply KMs on larger models in my opinion.
2. I think the authors should conduct some analysis on why distiling under the DDCD setting is better than continual pre-training on the new data, since DDCD itself is essentially a next-word prediction objective, with the sole difference being mimicing hard targets (word IDs) or teacher model logits. I think the intuition and potential explanation behind this can be applied and beneficial to many other relevant research directions.

---

> ### Author Response · Authors · 2025-05-31
> **Response to LZaV**
>
> Thank you for carefully and thoroughly reviewing our submission, and for the constructive feedback provided. We address the points you raised below.
>
> $\texttt{ }$
> ### On the relative performance of KMs w.r.t ICL/RAG
>
> We agree with the reviewer that practitioners should use ICL / RAG if they can afford the added inference cost of these methods. Even in such scenarios, KMs can yield additional gains when applied in conjunction with ICL / RAG, effectively empowering users to boost performance across a wide range of inference budgets (see Fig. 2). In any case, we agree that studying the scaling properties of knowledge modules with larger base models would complement the current analysis; we leave this for future work.
>
> $\texttt{ }$
> ### On the surprising effectiveness of DDCD vs LM
>
> Thank you for highlighting this. We believe that the superiority of DDCD comes from the richer signal that the distillation objective provides. Indeed, both the hidden state matching loss, as well as the KL objective, yield more information to the learner than the standard next-token prediction objective. This result aligns with the general finding that distillation (teacher model matching) is more data-efficient than NTP (data matching) in the literature [3-4].
>
> To prove this, we run the following experiment: We progressively strip components of DDCD, making it closer to the LM baseline; we remove the l1 loss on the hidden states, and progressively lower the temperature $\tau$ when computing the teacher probabilities $p_{teacher} = \text{softmax}(\frac{logits}{\tau})$, until
> The target distribution becomes one-hot, effectively mimicking the properties of next-token-prediction loss.
>
> | Model | Rouge-L |
> |--|--:|
> | DDCD | 21.2 |
> | DDCD without hidden state loss | 19.0 |
> | DDCD without hidden state loss, temp=0.1 | 16.2|
> | DDCD without hidden state loss, temp=0.001 | 16.0|
> | DDCD without hidden state loss, temp=0 | 15.9 |
> | | |
> | LM baseline | 15.2 |
> | | |
>
>
> From these results, we can see that the performance correspondingly converges to that of LM. Experiments are run on the NarrativeQA dataset with phi-3 as the base model. Thus, we conclude that the overall distillation objective is central to the observed gains w.r.t LM.
>
>
> [3] He, Ruifei, et al. "Knowledge distillation as efficient pre-training: Faster convergence, higher data-efficiency, and better transferability."
>
> [4] Minixhofer, et al. “Universal Cross-Tokenizer Distillation via Approximate Likelihood Matching”

---

> > ### Comment · Reviewer_LZaV · 2025-06-01
> > **Thanks for the commments**
> >
> > Thanks to the authors for providing the comments. They are helpful and confirm that my understanding of the work is fair and correct. As a result, I believe my current evaluation is sound and reflects the general merit of the paper, and I will keep my current score.

---

### Official Review · Reviewer_ndgn · 2025-05-15

**Rating:** 7
**Confidence:** 3
**Ethics Flag:** 1

**Summary:**

- This study introduces a new approach for integrating new knowledge into language models through a modular, plug-and-play framework based on knowledge distillation with next-token prediction. Experiments across standard benchmarks demonstrate the effectiveness of the proposed method.

**Reasons To Accept:**

- The methodology is both theoretically sound and practically compelling, presenting a solution that is likely to generate substantial interest within the research community.

**Reasons To Reject:**

- It would benefit from further experiments focusing on potential negative side effects associated with knowledge injection, as reported in existing knowledge editing literature.

---

> ### Author Response · Authors · 2025-05-31
> **Response to ndgn**
>
> Thank you for reviewing our submission.
>
> We believe our approach to be more robust to downsides associated with knowledge injection, given the plug-and-play nature of Knowledge Modules. Indeed, if other information is altered in unintended ways, we always have the opportunity to downweight (or outright remove) the Knowledge Module for a given document. Because we never modify the base model, we can always revert to a previous capability present in the base model. That said, we test how much general knowledge is forgotten after Knowledge Module training, per your suggestion. We evaluate the KMs on the MMLU dataset, where for each datapoint, we randomly sample a KM with which to answer the question. We use a subset of 400 examples, evenly split across all MMLU subjects. We compare to the base model, llama 3 8B-Instruct below :
>
> | | |
> |--|--:|
> |**Model** | **Accuracy** |
> |base model | 63.9 |
> |base model + KM | 62.9 |
>
> We see that the base model equipped with a Knowledge Module is able to retain its overall general knowledge, dropping only 1 point on MMLU. We will add this experiment to the camera-ready version.
>
> Thank you again for your suggestion!

---

> > ### Comment · Reviewer_ndgn · 2025-06-11
> >
> > Thank you for your clarification and for sharing the additional experiments. As mentioned earlier, this paper presents findings that are likely to be of interest to this community. After reviewing the discussions with other reviewers as well, I have decided to raise my rating.

---

### Official Review · Reviewer_HsaC · 2025-05-17

**Rating:** 7
**Confidence:** 4
**Ethics Flag:** 1

**Summary:**

In this paper, the authors introduce knowledge module, a lightweight adapter that encode document-level knowledge into continuous parameters that can be plugged into pretrained LLMs for improved knowledge-intensive tasks. They proposed deep context distillation (DCD) is a novel training paradigm that distills both hidden representations and output distributions from a teacher model with access to the document. To improve training, the authors generate synthetic data in the form of summaries, QA pairs etc., demonstrating that DCD can outperforms baselines on long-context QA tasks across close and open-book settings. The authors also introduce task-specific knowledge extractors to combine KMs with retrieval-augmented generation, with additional gains and highlighting the effectiveness of the proposed method.

**Reasons To Accept:**

(1) The authors propose a very interesting approach to distill context into parametric knowledge for improved knowledge-intensive tasks

(2) The authors show different design choices of the knowledge modules and knowledge extractors, providing insights on how these can be optimally combined for improved performance

(3) The proposed KM + KE works quite well on the adopted datasets and could also be used for RAG, in which the documents are retrieved and then the corresponding adaptors can be loaded for the LLMs

**Reasons To Reject:**

(1) The proposed method might be too expensive to train for large amounts of documents

(2) Although the proposed method is intriguing, I doubt this can scale for any existing corpus like Wikipedia and on more commonly used open-domain QA datasets like HotpotQA, 2Wiki etc. Therefore a more detailed analysis and comparison to existing RAG / encoding / efficient retrieval might improve the presentation of the mehtod.

---

> ### Author Response · Authors · 2025-05-31
> **Response to HsaC**
>
> Thank you for your detailed review of our work. We will address the concerns you raised point by point.
>
> $\texttt{ }$
> ### On the training cost for Knowledge Modules
> We kindly refer you to the global response, in which we compare the training cost of KMs to the inference cost of ICL and RAG. We show that in settings where many queries are performed for the same document (e.g., an enterprise chatbot answering questions about company resources), the training cost is quickly amortized over a short timespan.
>
> $\texttt{ }$
> ### Scaling to other datasets & more detailed analysis
> Regarding scaling to datasets such as HotpotQA and 2Wiki, we believe our approach could be deployed in such settings, as our synthetic data generation step would help expand their relatively small documents.
>
> Regarding scaling to the full Wikipedia collection, one could train a KM for each Wikipedia page. Since in open-domain settings the relevant pages are unknown, one could use 0-shot routing strategies [2] to determine on-the-fly which KM to use. This is a very interesting research direction, and we leave this for future work, along with a more detailed analysis and comparison of different types of data formats and methods.
>
> Thank you again for the positive feedback!
>
> [2] Muqeeth, Mohammed, et al. "Learning to route among specialized experts for zero-shot generalization."

---

> > ### Comment · Reviewer_HsaC · 2025-06-09
> >
> > Thanks for the response, I will keep my score as is.

---

### Author Response · Authors · 2025-05-31
**General Response**

We thank the reviewers for their overall positive outlook on our work and for their constructive feedback.  Two reviewers raised questions regarding the training cost of Knowledge Modules, which we address below.

$\texttt{ }$
### On the cost of training Knowledge Modules

We generally find that the cost of training KMs is proportional to the length of said documents; longer documents require more training steps. That said, longer documents are also costlier to run in-context; Indeed, for our experiments on NarrativeQA, we find that the full cost of training KMs in FLOPS is equivalent to **173 ICL inference queries**. Therefore, the training cost is amortized over time as the number of inference queries rises. Importantly, KM training *shifts the cost incurred from the user to the developer*, making the user experience better as it reduces latency, a property which we believe to be very important in many settings.

We will add a section on the cost-related matters into the camera-ready version, following the reviewers’ feedback.

$\texttt{ }$
### Cost breakdown

We use the approximation in [1], where a decoder-only Transformer with N parameter uses 2N FLOPS per token at inference and 6N FLOPS per token for training. For training of NarrativeQA with (at most) 1500 training steps, where the teacher sequence length is 4096 and the mean student sequence length is 942 tokens, the total cost is

`n_training_steps x ( student_cost + teacher_cost) =
1500 x (6N * 942 + 2N * 4096) = 20.8M x N FLOPs`

On the other hand, a single ICL query with the full document (assuming an average sequence length of 60000 tokens) gives:

`2N * 60000 = 0.12M x N FLOPs`

Thus, after `20.8 / 0.12 = 173` inference calls, the training cost is fully amortized.

We also compute this equivalence for RAG. Without counting the cost to embed the document chunks, and the cost to perform the retrieval, when performing inference calls with RAG, we typically use the full 4K context window. This gives us a cost of:

`2N * 4096 = 8.192K x N FLOPs`

Thus, the training cost is amortized after `20.8M / 8.192K  = 2539` inference calls.

[1] Jared Kaplan, Sam McCandlish, T. J. Henighan, Tom B. Brown, Benjamin Chess, Rewon Child, Scott Gray, Alec Radford, Jeff Wu, and Dario Amodei. Scaling laws for neural language models.

---

### Decision · Program_Chairs · 2025-07-08

**Decision:**

Accept

**Comment:**

The paper proposes a creative approach to injective new knowledge into LLMs using LoRA adapters (called Knowledge Modules) trained via distillation from a teacher LLM. The reviewers broadly find this paper exciting, particularly highlighting the (1) plug-and-play nature of the approach, (2) potential use cases in private domains.

The authors should address the following points to strengthen their next version of the paper:

1. (Reviewer LZaV) Conduct addtitional analysis on why distillation is superior to simple comtinued pre-trianing on the new data. Additionally, include results from prior continued pre-training baselines such as “Synthetic continued pretraining” (https://arxiv.org/abs/2409.07431) that report superior results on Quality.
2. Include discussion of the current approach’s limitations in scaling up the approach to large datasets.

[Automatically added comment] At least one review was discounted during the decision process due to quality]